# Contextual HyperNetworks for Novel Feature Adaptation

## Abstract

While deep learning has obtained state-of-the-art results in many applications, the adaptation of neural network architectures to incorporate new output features remains a challenge, as a neural networks are commonly trained to produce a fixed output dimension. This issue is particularly severe in online learning settings, where new output features, such as items in a recommender system, are added continually with few or no associated observations. As such, methods for adapting neural networks to novel features which are both time and data-efficient are desired. To address this, we propose the Contextual HyperNetwork (CHN), an auxiliary model which generates parameters for extending the base model to a new feature, by utilizing both existing data as well as any observations and/or metadata associated with the new feature. At prediction time, the CHN requires only a single forward pass through a neural network, yielding a significant speed-up when compared to re-training and fine-tuning approaches. To assess the performance of CHNs, we use a CHN to augment a partial variational autoencoder (P-VAE), a deep generative model which can impute the values of missing features in sparsely-observed data. We show that this system obtains improved few-shot learning performance for novel features over existing imputation and meta-learning baselines across recommender systems, e-learning, and healthcare tasks.

## 1 Introduction

In many deep learning application domains, it is common to see the set of predictions made by a model grow over time: a new item may be introduced into a recommender system, a new question may be added to a survey, or a new disease may require diagnosis. In such settings, it is valuable to be able to accurately predict the values that this feature takes within data points for which it is unobserved: for example, predicting whether a user will enjoy a new movie in a recommender system, or predicting how a user will answer a new question in a questionnaire.

On the introduction of a new feature, there may be few or no labelled data points containing observed values for it; a newly added movie may have received very few or even no ratings. The typically poor performance of machine learning models in this low-data regime is often referred to as the *cold-start* problem (Schein et al., 2002; Lika et al., 2014; Lam et al., 2008), which is prevalent not only in recommender systems but also in applications where high quality data is sparse. This presents a key challenge: the adaptation of a deep learning model to accurately predict the new feature values in the low data regime. On one hand, it is often required to deploy the model in applications immediately upon the arrival of new features, so it is impractical for the adaptation to wait until much more data has been acquired. On the other hand, simply retraining the model every time a new feature is introduced is computationally costly, and may fall victim to severe over-fitting if there are only a small number of observations available for the new feature.

Few-shot learning (Snell et al., 2017; Requeima et al., 2019; Vinyals et al., 2016; Gordon et al., 2018) has seen great successes in recent years, particularly in image classification tasks; however, these approaches typically treat all tasks as independent of one another. We wish to extend these ideas to the challenge of extending deep learning models to new output features, using a method which captures how a new feature relates to the existing features in the model. Furthermore, we seek a method that is computationally efficient, ideally requiring no fine-tuning of the model, and that is resistant to over-fitting in the few-shot regime.

To address these needs simultaneously, our contributions are as follows:

- *We propose an auxiliary neural network, called Contextual HyperNet (CHN), that can be used to initialize the model parameters associated with a new feature (see Section 2).*
  CHNs are conditioned on both a *context set* made up of observations for the new feature, and any associated content information or *metadata*. CHNs amortize the process of performing gradient descent on the new parameters by mapping the newly observed data directly into high-performing new parameter values, with no additional fine-tuning of the model being required. This makes CHNs highly computationally efficient and scalable to large datasets.
- *We use a CHN to augment a partial variational autoencoder (P-VAE) and evaluate the system's performance across a range of applications (see Section 4).*
  While CHNs are applicable to a wide range of deep learning models, in this work we choose a P-VAE as the evaluation framework. The result is a flexible deep learning model able to rapidly adapt to new features, even when the data is sparsely-observed, e.g. in recommender systems. We show that this model outperforms a range of baselines in both predictive accuracy and speed across recommender system, e-learning and healthcare tasks.

## 2 MODEL

### 2.1 PROBLEM SETTING

Our goal is to enable fast adaptation of a machine learning model when new output features are added to augment the originally observed data. Specifically, we consider the original observations as a set of vector-valued data points $\mathcal{D} = \{\mathbf{x}^{(i)}\}_{i=1}^{m}$, where each of the feature values in a given data point may be missing. We denote $x_j$ as the $j$th feature of a data point $\mathbf{x} \in \mathcal{D}$ and group the observed and unobserved features within a data point as $\mathbf{x} = [\mathbf{x}_O, \mathbf{x}_U]$. In many scenarios, such as recommender systems, a machine learning model $p(\mathbf{x}_U|\mathbf{x}_O)$ aims then at predicting the unobserved features $\mathbf{x}_U$ given observed ones $\mathbf{x}_O$.

Now suppose a new output feature $x_n$ becomes available, so that each data vector $\mathbf{x} \in \mathcal{D}$ is augmented to become $\tilde{\mathbf{x}} = [\mathbf{x}; x_n]$. This happens when e.g. a new item is added to a recommender system, or a new type of diagnostic test is added in a medical application. We note that not every data point $\mathbf{x} \in \mathcal{D}$ receives an *observed* value for the new feature: a newly added movie may have received very few ratings, or a new diagnostic test may have yet to be performed on all of the patients. We refer to the set of data points where the new feature is *observed* as the *context set* for the new feature $n$, i.e.

$$\mathcal{C}_n = \{\tilde{\mathbf{x}} = [\mathbf{x}; x_n] \mid \mathbf{x} \in \mathcal{D}, x_n \text{ is observed}\}.$$

context set is shown in yellow in Figure 1 Its complement, the *target set* $\mathcal{T}_n$, is the set of those data points for which there is no associated observation for the feature:

$$\mathcal{T}_n = \{\tilde{\mathbf{x}} = [\mathbf{x}; x_n] \mid \mathbf{x} \in \mathcal{D}, x_n \text{ is unobserved}\}.$$

Figure 1: Data used when adapting to a new feature $x_n$. The rows which contain yellow blocks are $\mathcal{C}_n$ and other rows are in $\mathcal{T}_n$.

One can also split the augmented data into observed and unobserved parts, i.e. $\tilde{\mathbf{x}} = [\tilde{\mathbf{x}}_O, \tilde{\mathbf{x}}_U]$. Using this notation, it is clear that $\tilde{\mathbf{x}}_O = [\mathbf{x}_O; x_n], \tilde{\mathbf{x}}_U = \mathbf{x}_U$ for $\tilde{\mathbf{x}} \in \mathcal{C}_n$, and $\tilde{\mathbf{x}}_O = \mathbf{x}_O, \tilde{\mathbf{x}}_U = [\mathbf{x}_U; x_n]$ for $\tilde{\mathbf{x}} \in \mathcal{T}_n$. In addition, we may also have access to some *metadata* $\mathcal{M}_n$ describing the new feature. This could be categorical data such as the category of a product in a recommender system or the topic of a question in an e-learning system, or some richer data format such as images or text.

We wish to adapt the machine learning model $p_{\boldsymbol{\theta}_0}(\mathbf{x}_U|\mathbf{x}_O)$ to $p_{\boldsymbol{\theta}}(\tilde{\mathbf{x}}_U|\mathbf{x}_O)$ so that it is able to accurately predict the value of the unobserved new features for data points $\tilde{\mathbf{x}} \in \mathcal{T}_n$. A naive strategy would ignore the previous model $p_{\boldsymbol{\theta}_0}(\mathbf{x}_U|\mathbf{x}_O)$ and instead seek the maximum likelihood estimates (MLE) of the parameters for the new model $p_{\boldsymbol{\theta}}(\tilde{\mathbf{x}}_U|\mathbf{x}_O)$. This is typically done by training the new model on the context set, by temporarily moving the observed new features $x_n$ to the prediction targets:

$$\hat{\boldsymbol{\theta}} = \arg\max_{\boldsymbol{\theta}} \left[ \sum_{\tilde{\mathbf{x}} \in \mathcal{C}_n} \log p_{\boldsymbol{\theta}}\left(x_n, \mathbf{x}_U|\mathbf{x}_O\right) \right].$$

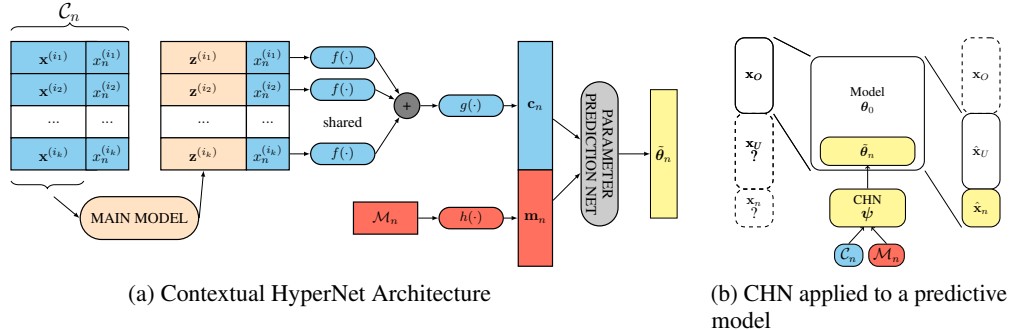

(a) Contextual HyperNet Architecture

(b) CHN applied to a predictive model

Figure 2: (a) Contextual HyperNetwork architecture. $\mathbf{z}^{(i)}$ is a fixed-length internal representation of the previously observed values in the data point $\mathbf{x}^{(i)}$ taken from the base model. (b) Complete architecture when a Contextual HyperNetwork is used to augment a predictive model $p_{\boldsymbol{\theta}_O}(\mathbf{x}_U|\mathbf{x}_O)$ to be able to make predictions for a new feature $n$, by initialising new parameters $\tilde{\boldsymbol{\theta}}_n$.

However, in deep neural networks, the number of model parameters $\boldsymbol{\theta}$ may be extremely large, so that maximising this log-likelihood is very expensive, particularly if new features are being introduced on a regular basis. Furthermore, optimising $\boldsymbol{\theta}$ for one particular feature may lead to poor performance for another, as is the case in *catastrophic forgetting* (Kirkpatrick et al., 2017) in continual learning tasks. In order to address both of these concerns, we divide the model parameters into parameters $\boldsymbol{\theta}_0$ inherent from the old model, and feature-specific parameters $\boldsymbol{\theta}_n$ associated solely with the new feature. In other words, we use $p_{\boldsymbol{\theta}_0}(\mathbf{x}_U|\mathbf{x}_O)$ as a base model and pose a factorisation assumption on the augmented model as $p_{\boldsymbol{\theta}}(\tilde{\mathbf{x}}_U|\mathbf{x}_O) = p_{\boldsymbol{\theta}_0}(\mathbf{x}_U|\mathbf{x}_O)p_{\boldsymbol{\theta}_0}(x_n|\mathbf{x}_O;\boldsymbol{\theta}_n)$, which together yield a predictive model for the new feature. We then hold $\boldsymbol{\theta}_0$ fixed and only seek MLEs for $\boldsymbol{\theta}_n$. While this greatly reduces the dimensionality of the parameter space over which we optimize for a new feature, and decouples the optimization of parameters for one new feature from another, several issues still exist. This factorization still requires a gradient descent procedure, which can be computationally costly and risks severe overfitting when there is little data for the new feature. Furthermore, it is not immediately clear how to make the estimation of $\boldsymbol{\theta}_n$ depend on the feature metadata $\mathcal{M}_n$. To address these problems, we introduce a *Contextual HyperNetwork* (CHN) $H_{\boldsymbol{\psi}}(\mathcal{C}_n, \mathcal{M}_n)$, an auxilliary neural network that amortizes the process of estimating $\boldsymbol{\theta}_n$. The goal is that when a new feature $x_{n^*}$ is added at test time, the CHN will directly generate "good" parameters $\tilde{\boldsymbol{\theta}}_{n^*} = H_{\boldsymbol{\psi}}(\mathcal{C}_{n^*}, \mathcal{M}_{n^*})$ such that the new predictive model $p_{\boldsymbol{\theta}_0}(x_{n^*}|\mathbf{x}_O;\boldsymbol{\theta}_{n^*} = \tilde{\boldsymbol{\theta}}_{n^*})$ can predict the values of the new feature accurately.

## 2.2 CONTEXTUAL HYPERNETWORKS

CHNs aim to map the context set $\mathcal{C}_n$ and metadata $\mathcal{M}_n$ into an estimate of the new model parameters $\tilde{\boldsymbol{\theta}}_n$. Since the size of $\mathcal{C}_n$ is variable for each feature, CHNs require an architecture that is able to adapt to a varying input dimension. This challenge is addressed through the use of a PointNet-style set encoder (Qi et al., 2017; Zaheer et al., 2017). For each context point $\tilde{\mathbf{x}}^{(i)} \in \mathcal{C}_n$, we concatenate the new feature $x_n^{(i)}$ with a fixed-length encoding $\mathbf{z}^{(i)}$ (see below) of the other observed features $\mathbf{x}_O^{(i)}$ within the data point. Each of these concatenated vectors $[\mathbf{z}^{(i)}, x_n^{(i)}]$ is then input to a shared neural network $f(\cdot)$, and the outputs $f([\mathbf{z}^{(i)}, x_n^{(i)}])$ are aggregated by a permutation-invariant function such as summation in order to produce a single, fixed-length vector. This output is passed through a second neural network $g(\cdot)$ to produce a fixed-length encoding of $\mathbf{c}_n$ we term a "context vector". This architecture is displayed in Figure 2a.

The fixed length encoding $\mathbf{z}^{(i)}$ of the observed features $\mathbf{x}_O^{(i)}$ for each context data point $\tilde{\mathbf{x}}^{(i)} \in \mathcal{C}_n$ is obtained by inputting the observed features to the base model $p_{\boldsymbol{\theta}_0}(\mathbf{x}_U|\mathbf{x}_O)$ and taking some intermediate representation from within the model: in an autoencoder model, this could be the encoded vector representing the data point at the information bottleneck, while in a feed-forward model it could be the output of an intermediate layer. By encoding the existing features in this way, we hope to enable the CHN to interpret the observed values for the new features *in the context of the base model's representation of remainder of the data point*.

Additionally, any feature metadata $\mathcal{M}_n$ is passed through a neural network $h(\cdot)$ to produce a fixed length metadata embedding vector $\mathbf{m}_n$. In the case of image or text metadata, specialized architectures

such as convolutional or sequence models can be used here. The concatenated vector $[\mathbf{c}_n; \mathbf{m}_n]$ is then input into a final feed-forward neural network which outputs the new feature-specific parameters $\tilde{\boldsymbol{\theta}}_n$. CHNs can be applied to any predictive model with dynamically added output features: Figure 2b shows the application of CHN to an predictive model. As an example, we illustrate how CHN is used with an autoencoder-style model in Figure 3.

Since it is possible to parallelize the encoding of the context set $\mathcal{C}_n$ both across each data point $\mathbf{x}^{(k)}$ and across the observed features within each data point $x_i$, the computational costs for parameter prediction with CHNs are able to scale efficiently with both the context set size and the number of observed values within each data point in the context set. When combined with the lack of any costly iterative gradient-descent procedure, this makes CHNs an extremely efficient choice for parameter initialization.

Figure 3: Contextual HyperNetwork applied to a P-VAE. The CHN generates parameters $\tilde{\boldsymbol{\theta}}_n$ for a new *decoder head*.

## 2.3 TRAINING CHNS WITH META-LEARNING

We adopt a *meta-learning* approach to training the CHN, treating the prediction of the values of each new feature as an individual task, with the aim of producing a model that can "learn how to learn" from $\mathcal{C}_n$ and $\mathcal{M}_n$. First, a base model $p_{\boldsymbol{\theta}_0}(\mathbf{x}_U|\mathbf{x}_O)$ is trained on the data observed before the adaptation stages; this model is then frozen during CHN training. To implement the training strategy, in the experiments we divide the dataset into three disjoint sets of features (see Figure 4): a 'training' set for base model training in the first stage, a 'meta-training' set for CHN meta-learning in the second stage, and a meta-test set for CHN evaluation in the third stage.

**Meta-Training of the CHN**  During meta-training, the parameters $\boldsymbol{\theta}_0$ of the base model are frozen, and we learn the parameters $\boldsymbol{\psi}$ of the CHN. We iterate the following training steps in mini-batches of features $\mathcal{B}$ sampled from the meta-training set for every step:

1. For each feature $n$ in $\mathcal{B}$, sample $k_n$ data points in which this feature is observed to form the context set $\mathcal{C}_n$, and reveal the associated feature values to the model. In our experiments we sample $k_n \sim \text{Uniform}[0, ..., 32]$ to ensure that a single CHN can perform well across a range of context set sizes.

2. For each feature $n \in \mathcal{B}$, compute feature-specific parameter predictions using the CHN,

$$\tilde{\boldsymbol{\theta}}_n = H_{\boldsymbol{\psi}}(\mathcal{C}_n, \mathcal{M}_n).$$

3. For each feature $n \in \mathcal{B}$, estimate the log-likelihood of the CHN parameters $\boldsymbol{\psi}$ given the ground truths for the hidden values of the feature $n$ in the data points in its target set $\mathcal{T}_n$, using the augmented model $p_{\boldsymbol{\theta}_0}(x_n|\mathbf{x}_O, \tilde{\boldsymbol{\theta}}_n)$:

$$l(\boldsymbol{\psi}) = \frac{1}{\sum_{n \in \mathcal{B}} |\mathcal{T}_n|} \sum_{n \in \mathcal{B}} \sum_{\{i|\mathbf{x}^{(i)} \in \mathcal{T}_n\}} \log p_{\boldsymbol{\theta}_0}(x_n^{(i)} \mid \mathbf{x}_O^{(i)}; \tilde{\boldsymbol{\theta}}_n).$$

4. Update the CHN parameters by taking a gradient ascent step in $\boldsymbol{\psi}$ for $l(\boldsymbol{\psi})$.

Note that the log-likelihood is only computed for the *hidden* values of the new feature in the target set $\mathcal{T}_n$, and not for the observed values in $\mathcal{C}_n$ – this is to ensure that the CHN produces parameters which generalize well, rather than overfitting to the context set. This approach is consistent with many meta-learning methods such as MAML (Finn et al., 2017), where the meta-learning model is updated based on its performance on a "test set" of previously unseen examples for each new task.

**Evaluating the CHN**  At evaluation time, the parameters of both the base model and the CHN are now frozen. A fixed context set and metadata are provided for each feature in the meta-test set, and

these are used to initialize feature-specific parameters for the meta-test features using the trained CHN. These parameters are then used to make predictions for all of the target set values for the new features, from which evaluation metrics are computed.

## 3 RELATED WORK

CHNs aim to solve the problem of adapting to a new feature with very few available observations, and thus relate to few-shot learning and related fields such as meta-learning and continual learning. From a technical point of view, we use an auxiliary neural network to amortize the learning of parameters associated with the new feature, which falls under the domain of hypernetworks. Furthermore, in the context of recommender systems, a number of related methods have been proposed to address the cold-start problem. We thus discuss related work in these three areas.

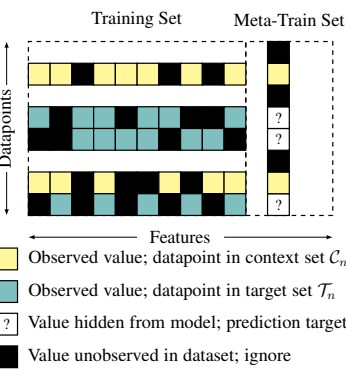

Figure 4: Data splits when meta-training a CHN. Meta-testing proceeds analogously, using features from an additional meta-test set of features, but using a constant set of observed values for each feature on every meta-test set evaluation.

**Few-Shot Learning**  Few-shot learning is the problem of designing machine learning models that can adapt to new prediction tasks given a small number of training examples. A popular approach to this problem is gradient-based meta-learning, such as MAML (Finn et al., 2017) and Reptile (Nichol & Schulman, 2018), which seek a parameter initialisation $\boldsymbol{\theta}$ that can rapidly adapt to tasks drawn from a task distribution $p(\mathcal{T})$. These methods do not directly condition the parameter initialisation for a new task on any associated data or metadata, instead relying on fine-tuning, which can be both computationally expensive and lead to severe overfitting when little data is available. Another line of methods seek to adapt a classifier to a task based on a context set of class examples. For instance, by embedding class examples to provide a nearest neighbours classifier (Snell et al., 2017), learning an attention mechanism between class examples and a new example (Vinyals et al., 2016), or modulating activation functions (Munkhdalai et al., 2017) within a feature extractor conditioned on the context set. Conditional Neural Adaptive Processes (Requeima et al., 2019), which are based on Conditional Neural Processes (Garnelo et al., 2018), adapt both classifier and feature extractor parameters based on the context set for a task. Similarly, Gidaris & Komodakis (2018) generate classifier weights for a new image class based on features extracted using the base model. However, in all cases, each task or image class is treated as independent from all others, whereas CHNs explicitly utilize all previously-observed features in the base model when adapting to a new feature.

A closely related field is continual learning (Kirkpatrick et al., 2017; Nguyen et al., 2017), where a model seeks to adapt to new tasks or a shifting data distribution while avoiding catastrophic forgetting of old tasks. Continual learning does not necessarily address the few-shot scenario and is commonly applied in classification settings where the classifier/heads can be either shared or independent. CHNs can be seen as a means of addressing continual learning in the few-shot learning regime, by generating parameters for a new feature conditioned on all of the features already learned by the model.

**Hypernetworks**  Hypernetworks (Ha et al., 2016) are auxiliary neural networks which generate the parameters of a neural network. They were introduced in Ha et al. (2016) as a form of model compression, with the hypernetwork taking as in input only structural information about the weight matrix they are currently predicting, such as its index in a sequence of layers. By contrast, CHNs are explicitly conditioned on data relevant to the weights currently being predicted. Bertinetto et al. (2016) train a hypernetwork to predict all of the parameters of a binary classifier for a class of images, conditioned on a single exemplar image for the class. It is found that the output dimension of the hypernetwork grows extremely large for even small classifiers—in order to mitigate this, the authors propose a factorisation of the parameters, whereas we instead choose to learn only a small number of feature-specific parameters $\boldsymbol{\theta}_n$. *Task-conditioned hypernetworks* (von Oswald et al., 2019) provide an application of hypernetworks to multi-task continual learning, where weights for the entire neural network for different tasks are predicted using a hypernetwork, based on a learned task embedding.

This setting differs from our work as the continual learning tasks are assumed to be independent, and the hypernetwork is not conditioned directly on data for the new task, instead requiring a gradient descent process to learn the associated task embedding.

**Cold Starts in Recommender Systems**   Cold starts (Schein et al., 2002; Lika et al., 2014) occur when there is little or no data associated with a novel item or user in a recommender system. *Collaborative filtering* approaches to recommender systems have enjoyed great success for many years (Schafer et al., 2007; Sarwar et al., 2001; He et al., 2017), but can fail completely when there is very limited rating data associated with a new user or item (Lam et al., 2008). One potential solution to cold starts is given by *content-based* methods (Pazzani & Billsus, 2007; Lops et al., 2011), which use any available descriptive information about the new user or item. Hybrid approaches (Balabanović & Shoham, 1997; Stern et al., 2009; Gomez-Uribe & Hunt, 2015) seek to marry these two approaches, making use of both collaborative and content-based methods. Meta-learning approaches also show promise for solving cold starts, including MAML-like approaches (Bharadhwaj, 2019) for initialising new items. Vartak et al. (2017) takes a similar approach to CHNs by adapting the weights of a classifier, but the adaption is based on user rather than item history, and it relies on class-specific item embeddings and thus is not extensible to regression settings. In this work, when applied to recommender systems, CHNs combine the strengths of all of these approaches, using content information, ratings data and latent representations of the associated users to generate accurate parameters for novel items. Recently, graph neural network (GNN)-based approaches (You et al., 2020; Schlichtkrull et al., 2018; Wang et al., 2019; Hamilton et al., 2017) have shown promise when applied to recommender systems. These methods formulate each item and each user as a node, and incorporate new items by adding nodes to the graph, where the graph structure may be exploited for inductive reasoning about new features. In this work, we target more standard, vector-based neural networks which are currently deployed across a wide range of application domains where GNNs may not be suitable, but future work may explore the application of CHNs to GNNs and the relationships between the two.

## 4   EXPERIMENTS

We demonstrate the performance of the proposed CHN in three different real-world application scenarios, including recommender systems (Section 4.2), healthcare (Section 4.3) and e-learning (Section 4.4). Our method exhibits superior performance in terms of prediction accuracy across all these applications. We also demonstrate an advantage in terms of computational efficiency in a large scale real-life setting.

### 4.1   EXPERIMENT SETTINGS

In all our experiments, we apply a CHN to a partial variational autoencoder (P-VAE) (Ma et al., 2018b;a) as an exemplar model. This is a flexible autoencoder model that is able to accurately work with and impute missing values in data points, allowing us to model sparsely-observed data such as that found in recommender systems. For each new feature $n$, we augment the P-VAE's decoder with a new *decoder head* consisting of an additional column of decoder weights $\mathbf{w}_n$ and an additional decoder bias term $b_n$ which extend the model's output to the new feature, so that $\boldsymbol{\theta}_n = \{\mathbf{w}_n, b_n\}$. Figure 3 illustrates how a CHN is used to extend a P-VAE to make predictions for a new feature $x_n$.

For all experiments, we train the CHN to output accurate feature parameters based on a range of context set sizes $k \in [0, ..., 32]$ by randomly sampling $k$ on each occurrence of a meta-training set feature. We then evaluate the performance of the CHN and baselines on the meta-test set features for a fixed range of context set sizes, ensuring that the same context sets are revealed to the CHN and each baseline. Further experimental results can be found in Appendix C, and full details on the model architectures and hyperparameters can be found in Appendix D. All results are averaged across 5 random train/meta-train/meta-test feature splits, and $\pm 1\sigma$ error bars are plotted across these splits. Hyperparameters and model architectures were tuned on different data splits to those used in the final experiments.

We consider the following baselines for generating the new feature parameters $\boldsymbol{\theta}_n = \{\mathbf{w}_n, b_n\}$. All methods are applied to the same base trained P-VAE model to ensure a fair comparison.

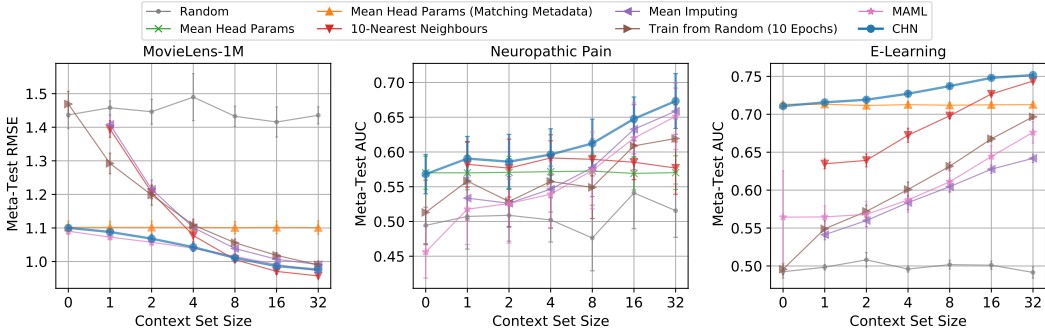

Figure 5: Performances ($\pm 1\sigma$) of CHN and benchmarks on MovieLens-1M (left), Neuropathic Pain (middle) and E-learning (right) datasets for test features, with varying context size $k$. We report meta-test RMSE for MovieLens-1M (lower is better) and AUROC for the others (higher is better). For the MAML baselines, the best-performing case is shown: 1 fine-tuning epoch for MovieLens-1M and 10 for others.

- **Random**: Generate a random value for $\boldsymbol{\theta}_n$ for each new decoder head using Xavier initialisation for weights and 0 for biases.
- **Mean Imputing**: Set weights and biases to always predict the mean of the observed values for the new feature in the context set, i.e. $\mathbf{w}_n = \mathbf{0}$ and $b = \sigma^{-1}\left(\frac{1}{k}\sum_{i\in\mathcal{C}_n} x_n^{(i)}\right)$.
- **Mean Head Parameters**: Generate the new head parameters $\boldsymbol{\theta}_n$ as the mean of all of the head parameters learned on the training set features.
- **Mean Head Parameters (Matching Metadata)**: As above, but averaging only over parameters of heads whose associated feature has metadata categories matching those of the new feature.
- **k-Nearest Neighbour Head Parameters**: Generate the new head parameters $\boldsymbol{\theta}_n$ as the mean of the head parameters of the $k$-nearest neighbour features in terms of Euclidean distance, where column-wise mean imputing is used to fill in unobserved values.
- **Train from Random**: Initialize the new feature head parameters randomly, and then fine-tune these parameters on the data in the context set $\mathcal{C}_n$ for a fixed number of epochs.
- **MAML**: We meta-learn an initialisation of $\boldsymbol{\theta}_n$ using Model-Agnostic Meta Learning (Finn et al., 2017), where we treat the prediction of each feature as a separate task and fine-tune these parameters on the context set. In all experiments, we compare with the MAML baseline which has the best-performing number of fine-tuning epochs. For full details, see Appendix B.

## 4.2 RECOMMENDER SYSTEMS

In real-life recommender systems, new users and new items are continuously added as new customers join and new products are launched. In deep learning based frameworks such as Sedhain et al. (2015); Liang et al. (2018); Gong et al. (2019); Ma et al. (2018a), the deep neural networks are commonly used in a user-based manner. In this approach, each new user is treated as a new data point, while each new item is treated as a new feature. To add a new item, one must extend the network architecture to incorporate the new feature, and we propose CHNs as an efficient way to predict the parameters associated with the new feature.

We evaluate the scenario above with MovieLens-1M dataset (Harper & Konstan, 2015). The dataset consists of 1 million ratings in the range 1 to 5 given by 8094 users to 5660 movies, and is thus 2.2% observed. For each movie, we have associated metadata $\mathcal{M}_n$ giving a list of genres associated with the movie, such as Action or Comedy, which we encode in a binary format, alongside the year of release which we normalize to lie within $[0, 1]$. For each random data split, we sampled 60% of movies as training data to train the base P-VAE model, used 30% as a meta-training set for CHN training and used the remaining 10% as a meta-test set.

The plot in Figure 5 (left) shows the performance of our proposed CHN, comparing with all other baselines in terms of RMSE (lower is better). Our method shows an advantage over all considered

baselines other than MAML in the few-shot regime ($k < 8$), while achieving competitive performance with MAML without requiring costly fine-tuning.

### 4.3 HEALTHCARE

In healthcare applications, a new question is often added to an existing health-assessment questionnaire, and in hospitals, new medical devices may be introduced to make physiological measurements. In this case it is desired for a model to quickly adapt to the newly added feature for health assessment, even when relatively few tests have been administered and so data is scarce.

We assess the utility of CHNs in a healthcare setting using synthetic data generated by the Neuropathic Pain Diagnosis Simulator (Tu et al., 2019). This simulator produces synthetic data using a generative model to simulate pathophysiologies, patterns and symptoms associated with different types of neuropathic pain. The data is binary, where a 0 represents the a diagnostic label that is not present in a patient's record, and a 1 indicates a diagnostic label that is present. We simulated 1000 synthetic patients, and removed features with fewer than 50 positive diagnoses, resulting in 82 remaining features, with 17.3% of the values in the dataset being positive diagnoses. We used $50\%$ of the features as training set; $30\%$ of the features as the meta-test set and $20\%$ of the features as the meta-test set.

The plot in Figure 5 (middle) shows the results in terms of AUROC (higher is better), as the dataset is highly imbalanced. Our method consistently outperforms all baselines across all values of $k$, while many methods including MAML suffer from severe overfitting when $k$ is small. In contrast to the MovieLens-1M result, here the 10-nearest neighbour approach does not seem to leverage more datapoints in the context set. This shows that our method is desirable in the cost-sensitive healthcare environment, even for highly imbalanced medical tests where results are largely negative.

### 4.4 E-LEARNING

We foresee CHNs being valuable in online education settings, potentially allowing teachers to quickly assess the diagnostic power of a new question given a small number of answers, or to gauge whether a new question's difficulty is appropriate for a particular student.

We assess the performance of the CHN in an e-learning setting using a real-life dataset provided by the e-learning provider Eedi for the NeurIPS 2020 Education Challenge (Wang et al., 2020). In particular, we use the dataset for the first 2 tasks, filtered so that all students and questions have at least 250 associated responses. This results in a dataset of for 6797 students across 4792 questions, detailing whether or not a student answered a particular question correctly. The dataset contains approximately 2.7 million responses, making it 8.2% observed. We treat each student as a data point and each question as a feature, and use a binary encoding of each question's associated subjects as metadata. We used $60\%$ of the questions as training set; $30\%$ of the questions as the meta-test set and $10\%$ of the questions as the meta-test set.

The right panel in Figure 5 illustrates the performance on prediction on the unseen meta-test set in terms of AUROC. Our method shows a significant improvement over all of the considered baselines over the entire range of $k$, suggesting promise for applying CHNs in educational settings.

### 4.5 TIMING EXPERIMENTS

We used the E-learning dataset to compare the time taken to generate new feature parameters at meta-test time for a number of methods considered in our experiments. The results are shown in Table 1. We see that the CHN offers nearly a 4-fold speedup compared to the nearest-neighbours based approach. We see a similar difference in performance when compared to training the new heads on a single observation for just 10 epochs. Moreover,

Table 1: Average time taken to initialize parameters for a feature in the e-learning dataset given $k$ observations. All times are given in milliseconds, averaged across the whole meta-test set using a batch size of 128. T Random here indicates Train from Random.

| Method/K | 1 | 4 | 16 |
|---|---|---|---|
| 10-NN | $400.8 \pm 0.5$ | $402.3 \pm 2.1$ | $405.1 \pm 1.5$ |
| T Random (1 Epoch) | $47.4 \pm 4.5$ | $61.3 \pm 4.7$ | $89.2 \pm 4.4$ |
| T Random (5 Epochs) | $210.9 \pm 20.4$ | $270.7 \pm 18.4$ | $331.1 \pm 20.0$ |
| T Random (10 Epochs) | $414.9 \pm 38.6$ | $530.3 \pm 35.9$ | $651.7 \pm 38.8$ |
| Contextual HyperNet | $113.7 \pm 1.1$ | $116.5 \pm 1.0$ | $119.9 \pm 1.3$ |

while this training time grows rapidly with
the number of observations in the context
set, the time taken for a CHN remains nearly constant since it can efficiently parallelize along these observations, making CHNs an extremely efficient initialisation choice for larger context set sizes. The experimental settings are specified in Appendix D where the total computation time for training and evaluating a CHN was approximately 3 minutes on the Neuropathic Pain dataset, 1.5 hours on the E-learning dataset, and 8 hours on MovieLens-1M.

## 5 CONCLUSION

We introduce Contextual HyperNetworks (CHNs), providing an efficient way to initialize parameters for a new feature in a model given a context set of points containing the new feature and feature metadata. Our experiments demonstrate that CHNs outperform a range of baselines in terms of predictive performance across a range of datasets, in both regression and classification settings, and are able to perform well across a range of context set sizes, while remaining computationally efficient. In the future work, we will evaluate CHNs in streaming setting with large-scale real-world applications.

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

# A    APPENDIX: PARTIAL VARIATIONAL AUTOENCODERS

## A.1    PARTIAL VARIATIONAL AUTOENCODERS

For our experiments, we base our model on the Partial Variational Autoencoder (P-VAE) (Ma et al., 2018a) - this model combines a traditional variational autoencoder (VAE) model with a PointNet-style set encoder (Qi et al., 2017), allowing it to efficiently encode and reconstruct partially observed data points. The P-VAE is based on the observation that typically the features in a VAE are assumed to be conditionally independent when conditioned on the latent variable $\mathbf{z}$. That is,

$$p(\mathbf{x}|\mathbf{z}) = \prod_j p(x_j|\mathbf{z})$$

Then, given a data point $\mathbf{x}$ with observed features $\mathbf{x}_O$ and unobserved features $\mathbf{x}_U$, we have that

$$p(\mathbf{x}_U|\mathbf{x}_O, \mathbf{z}) = p(\mathbf{x}_U|\mathbf{z})$$

Hence, if we can infer a posterior distribution over $\mathbf{z}$ from the observed features, we can use this to estimate $p(\mathbf{x}_U|\mathbf{x}_O)$. The P-VAE infers a variational posterior distribution over $\mathbf{z}$ using an amortized inference network (or *encoder* network) $q_\theta(\mathbf{z}|\mathbf{x}_O)$ and approximates the conditional data distribution given a value of $\mathbf{z}$ using a *decoder* network $p_\phi(\mathbf{x}_O, \mathbf{x}_U|\mathbf{z})$.

In our model, we extend the decoder to decode the value of a new feature $x_n$ by initialising an additional subnetwork in the decoder which we term a *decoder head*, with parameters $\phi_n$, to extend its output dimension by one. In principal this head could be of any architecture which takes as input the output of the shared layers of the decoder, but in practice we found that simply extending the final layer of weights and biases to accommodate a new output dimension yielded good results while remaining parameter-efficient as the number of output features grows.

## A.2    TRAINING P-VAEs

The P-VAE is trained to reconstruct observed features in the partially-observed data point, and in the process learn to infer a variational posterior $q_\theta(\mathbf{z}|\mathbf{x}_O)$ over the latent variable $\mathbf{z}$. The P-VAE is given batches of data points where features from both the meta-train and meta-test sets are hidden from the model. Additionally, each time a particular data point is input, some additional features are also randomly hidden from the model using a Bernoulli mask, in order to ensure the model is robust to different sparsity patterns in the data. The P-VAE is then trained by maximising the Evidence Lower-Bound (ELBO) (Ma et al., 2018b):

$$
\begin{aligned}
\log p(\mathbf{x}_O) &\geq \log p(\mathbf{x}_O) - D_{\mathrm{KL}}(q(\mathbf{z}|\mathbf{x}_O)||p(\mathbf{z}|\mathbf{x}_O)) \\
&= \mathbb{E}_{\mathbf{z} \sim q(\mathbf{z}|\mathbf{x}_O)} \left[\log p(\mathbf{x}_O, \mathbf{x}_U|\mathbf{z})\right] - D_{KL}\left[q(\mathbf{z}|\mathbf{x}_O)||p(\mathbf{z})\right] \\
&= \mathscr{L}_{partial}(\mathbf{x}_O)
\end{aligned}
$$

# B   APPENDIX: BASELINES

Here we provide additional details and results for the baselines used in our experiments.

## B.1   MAML

We adapt the Model-Agnostic Meta Learning (Finn et al., 2017) technique as a baseline. The decoder head parameters $\boldsymbol{\theta}_n$ are adapted using the MAML algorithm in the 'meta-training' stage. Each new feature $\mathcal{X}$ is viewed as a separate MAML task, with some observed and unobserved values. We sample the tasks in batches of size $M$ and train the inner (a.k.a. fast) model over $N$ steps. The inner model training loss is the ELBO of the PVAE on the observations $\mathcal{L}_{\mathcal{X}_O}$. The meta-model (a.k.a. the slow or outer model) is trained by being given the context set observations, and computing a reconstruction loss on the target set, $\hat{\mathcal{L}}_{\mathcal{T},\mathcal{C}}(f_{\theta'}, \mathcal{X})$. The gradient for the meta-model update is taken over the batch reconstruction losses mean. The full algorithm is detailed in Algorithm 1.

---

**Algorithm 1** Feature-wise Model-Agnostic Meta-Learning with PVAE

---

**Input:**
  $p(\mathcal{X})$: distribution over features.
  $\alpha, \beta$: learning rate hyperparameters.
  $M$: meta-batch size, $N$: number inner iterations.
Initialize $\theta$
**while** not done **do**
  Sample $M$ features $\mathcal{X}_i \sim p(\mathcal{X})$.
  **for all** $\mathcal{X}_i$ **do**
    $\theta_{i,0} \leftarrow \theta$
    **for** $j \leftarrow 0, N$ **do**
      Evaluate ELBO gradient $\nabla_{\theta_{i,j}} \mathcal{L}_{\mathcal{X}_O}(f_{\theta_{i,j}}, \mathcal{X}_i)$ w.r.t. observations in $K$ examples
      Optimize inner model parameters: $\theta_{i,j+1} \leftarrow \theta_{i,j} - \nabla_{\theta_{i,j}} \mathcal{L}_{\mathcal{X}_O}(f_{\theta_{i,j}}, \mathcal{X}_i)$
    **end for**
    $\theta'_i \leftarrow \theta_{i,N}$
  **end for**
  Evaluate gradient of mean reconstruction error $\nabla_\theta \frac{1}{M} \sum_{\mathcal{X}_i \sim p(\mathcal{X})} \hat{\mathcal{L}}_{\mathcal{T}_i, \mathcal{C}_i}(f_{\theta'_i}, \mathcal{X}_i)$
  Optimize meta-model parameters: $\theta \leftarrow \theta - \nabla_\theta \frac{1}{M} \sum_{\mathcal{X}_i \sim p(\mathcal{X})} \hat{\mathcal{L}}_{\mathcal{T}_i, \mathcal{C}_i}(f_{\theta'_i}, \mathcal{X}_i)$
**end while**
**Output:** $\theta$

---

Notably, since MAML aims to fit parameters that adapt quickly to new tasks, it allows for fine-tuning at evaluation time, that is, training the model for several iterations from the MAML parameter initialization. Here, we evaluate the model with and without fine-tuning.

In the MAML baseline experiments we use $M = 4$, $N = 10$, ADAM (Kingma & Ba, 2014) with learning rate $\alpha = \beta = 10^{-2}$ for inner and outer model optimization. The model fine-tuned performance is evaluated over $\{1, 3, 5, 10\}$ epochs and beset results are used. We make use of the higher order optimization facilitated by the `higher` library (Grefenstette et al., 2019) in the implementation of this baseline.

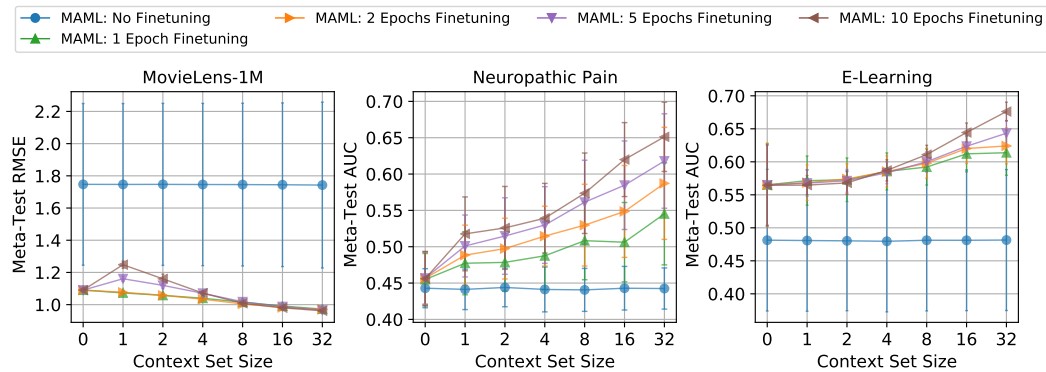

Figure 6: MAML baseline performance comparison for $\{1, 2, 5, 10\}$ fine-tuning epochs and with no fine-tuning. Left plot shows RMSE (lower is better), center and right plots show AUROC (higher is better).

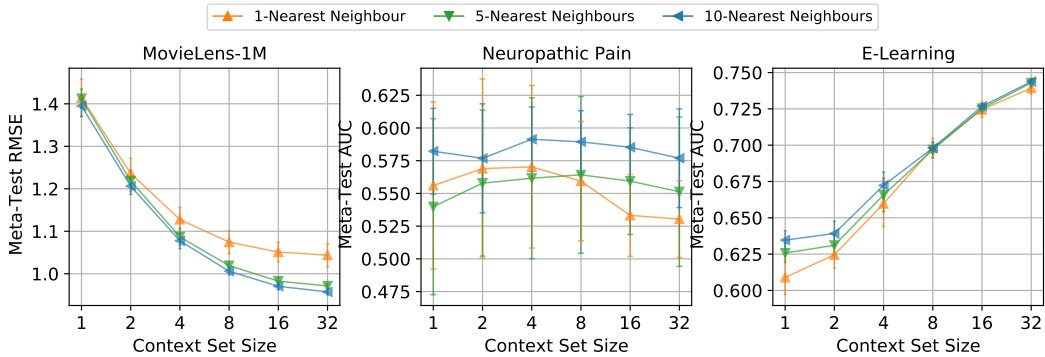

Figure 7: k-Nearest Neighbour Head Parameters baseline performance for $k \in \{1, 5, 10\}$. Context set size here corresponds to the number of observed values used when determining the nearest neighbour heads.

## C   APPENDIX: ADDITIONAL RESULTS

Here we provide additional results not included in the main text.

### C.1   MAML

Figure 6 shows the performance of the MAML baseline for different numbers of fine-tuning epochs and with no fine-tuning. As expected, the baseline with no fine-tuning is outperformed by those where fine-tuning is employed. For the Neuropathic Pain and E-learning datasets, the increase in the number of fine-tuning epochs corresponds to improvement in performance (greater AUROC), whereas in case of MovieLens-1M, performance drops (RMSE increases) with longer fine-tuning, particularly for the smaller context set sizes.

### C.2   k-NEAREST NEIGHBOUR HEAD PARAMETERS

We consider k-Nearest Neighbour Head Parameters baselines for the values $k \in \{1, 5, 10\}$. Figure 7 shows the performance of this baseline for the different values of $k$ across a range of context set sizes. We expect that as $k$ is increased further, and the number of head parameters averaged over grows, the behaviour will approach that of the mean head parameter baselines. In the main text, 10-Nearest Neighbours is used throughout, as it yields good performance in both the low and high-data regimes.

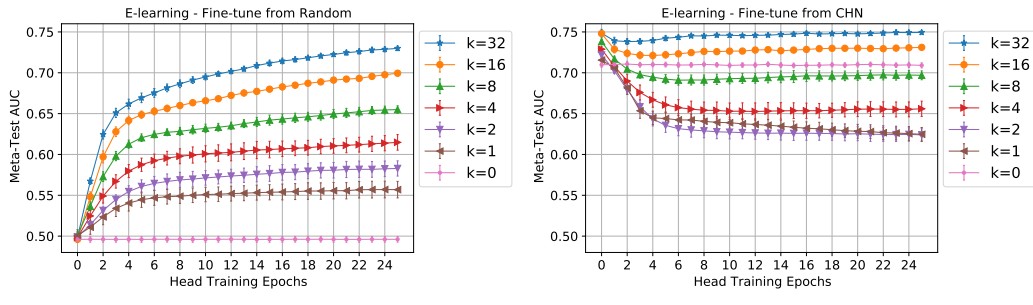

Figure 8: Comparing the predictive performance when training decoder new head parameters in a P-VAE on a range of context set sizes $k$, on the E-learning dataset (higher is better).

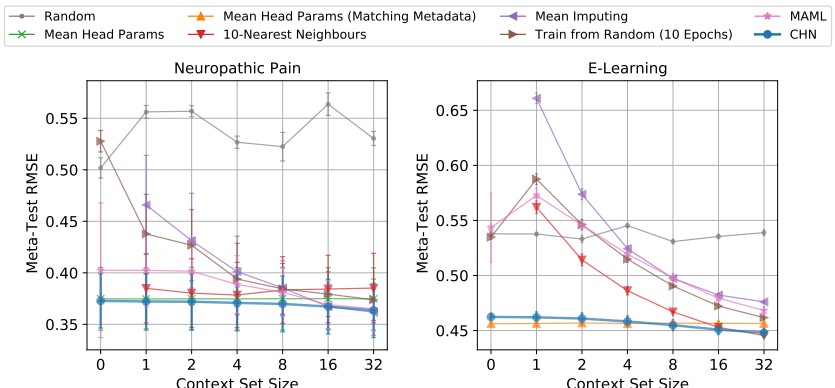

Figure 9: Performances ($\pm 1\sigma$) of CHN and benchmarks on Neuropathic Pain (left) and E-learning (right) datasets for test features, with varying context size $k$. We report RMSE for both datasets, in constrast to the main text where AUROC was reported.

## C.3 FINE-TUNING

In the main text, we show the performance of training the new decoder heads on their context sets from randomly initialized parameters for 10 epochs (see results for "Train from Random (10 epochs)"), in order to provide a trade-off between predictive accuracy and computational cost. This section aims to elaborate further on this trade-off. In Figure 8a, we show the predictive performance of the P-VAE on the meta-test set after training randomly initialized head parameters for an increasing number of epochs, for a range of context set sizes $k$. We see that the performance improves with training in all cases, with better performance achieved as the context set size $k$ increases, and thus the effect of over-fitting is lessened.

Furthermore, in Figure 8b, we perform the same experiment but instead initialising the heads with the CHN parameters. We see that in all cases except $k = 0$ and $k = 32$, training by gradient descent leads to a decrease in performance due to over-fitting, suggesting that the CHN has an implicit regularising effect on the parameter initialisation. We note also that in all cases, the untrained CHN parameters substantially outperform those trained from the random initialisation for all values of $k$, even after 25 training epochs, with many of the training curves appearing to approach convergence.

## C.4 RMSE VALUES FOR BINARY DATASETS

In Figure 9 we provide the RMSE values (lower is better) for the main paper experiments run on the Neuropathic Pain and E-learning datasets. We see that the relative performance ordering of methods in terms of RMSE is largely identical to the performance ordering obtained from the AUC results, suggesting that the proposed CHN approach is indeed better than other baselines.

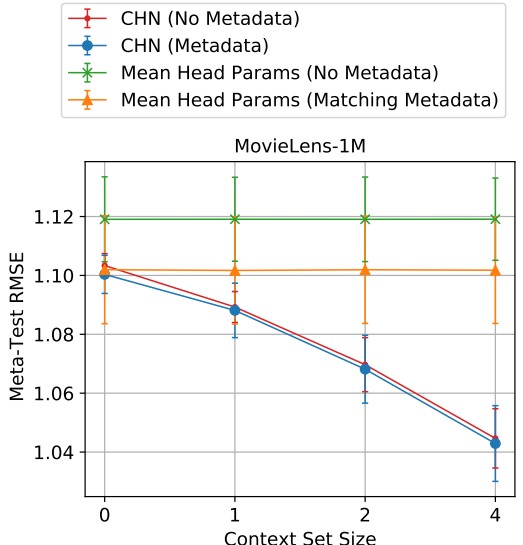

Figure 10: Comparison of RMSE (lower is better) with and without feature metadata for both the CHN and mean head parameter baselines, across 5 random data splits.

## C.5 METADATA ABLATION STUDY

It is possible to obtain an ablation for removing the usage of the context set $\mathcal{C}_n$ as an input to the CHN by looking at the performance for $k = 0$ in any of the preceding performance plots. In order to further investigate the effect of metadata on the CHN's performance, we additionally perform an ablation on MovieLens-1M by removing the use of metadata (movie genre and year) as an input to the CHN. We also compare the performance of averaging all trained head parameters, vs. averaging all trained head parameters whose features are a subset of those of the new feature under consideration. The results of these ablations can be found in Figure 10. We see that in both cases, in absolute terms the difference in performance due to the metadata is slight, with a small improvement made to the mean head parameter performance, and a very slight improvement to the CHN performance when $k = 0$. This is likely due to the metadata carrying very little information for this dataset relative to the data points themselves, so that when $k \geq 1$ the CHN is able to effectively disregard the metadata in favour of the observed data points. Future work could investigate the performance of CHNs on datasets with richer and more information feature metadata.

Table 2: Hyperparameters and architecture details for the P-VAE and CHN used on each dataset. Feed-forward neural networks are represented by a list of the dimensions of their hidden layers.

| | MovieLens-1M | Neuropathic Pain | E-learning |
|---|---|---|---|
| **Training** | | | |
| Epochs | 200 | 1000 | 50 |
| Batch Size | 1000 | 1000 | 1000 |
| Learning Rate | 1e-3 | 1e-2 | 1e-3 |
| Weight Decay | 0 | 0 | 0 |
| **Meta-Training** | | | |
| Epochs | 100 | 300 | 20 |
| Batch Size | 256 | 128 | 128 |
| Learning Rate | 1e-4 | 1e-3 | 1e-3 |
| Weight Decay | 1e-3 | 1e-3 | 1e-3 |
| **Set Encoder** | | | |
| Feature Embedding Dim. | 50 | 30 | 50 |
| Set Embedding Dim. | 30 | 30 | 30 |
| **Encoder** | | | |
| Latent Dim. | 150 | 20 | 150 |
| Layers | [200] | [30] | [200] |
| **Decoder** | | | |
| Shared Layers | [200] | [30] | [200] |
| Output Variance | 0.1 | - | - |
| **CHN** | | | |
| Data point Embedding Dim. | 50 | 25 | 50 |
| Context Encoding Dim. | 50 | 25 | 50 |
| Context Encoder Layers | [128] | [50] | [50] |
| Metadata Encoding Dim. | 5 | - | 20 |
| Metadata Encoder Layers | [10] | - | [20] |
| Param. Pred. Net Layers | [256,256,256] | [64,64] | [50,100,150] |

## D APPENDIX: EXPERIMENT DETAILS

All models were implemented in PyTorch (Paszke et al., 2017). All experiments were performed on a single Nvidia Tesla K80 GPU. For training both the P-VAE and the CHN's parameters, the ADAM (Kingma & Ba, 2014) optimizer was used with $\beta_1 = 0.9$, $\beta_2 = 0.999$ and $\varepsilon = 10^{-8}$

Details of hyperparameters and model architectures used for each dataset can be found in Table 2.

