# OpenReview forum: "Contextual HyperNetworks  for Novel Feature Adaptation"
_ICLR.cc/2021/Conference — Reject_

### Official Review · AnonReviewer3 · 2020-10-23
**Interesting work**

**Rating:** 6
**Confidence:** 2

**Review:**

This work proposes CHN, a framework to extend an existing model to incorporate new features as they become available. The benefits of CHN are demonstrated by utilizing it to extend a P-VAE.

This work proposes a structured procedure to incorporate new features in an incoming data for use with a trained model. The proposed CHN uses a combination of a context vector and a meta vector (generated through a neural network from the new features's meta data) to produce parameters $\hat{\theta}_n$ (for the new feature $x_n$). This $\hat{\theta}_n$ can then be used with the base model's representations (encoded vector for the P-VAE setup) for downstream tasks.

The work does seem to have merit as existence of missing raw features / new information about data in the new samples is not uncommon for practical scenarios. Therefore, an approach that does not require extensive resource usage to produce a new model that can utilize the new information is very useful.

This work empirically demonstrates the benefit of CHN over several tasks along with demonstrating its efficiency. There are, however, some points that need to be addressed.


In Figure 5, the x-axis is the context size which I am assuming is $k$. However, in section 2.3 it is mentioned that $k_n$ is sampled from Uniform[0,32]. In such a setup, what does it mean to have a fixed point for each k on the x-axis ? Does it mean that $k_n$ in this case was not sampled from a uniform distribution ? Perhaps Figure 5 was misunderstood by me and a clarification would be useful.


How is the new feature incorporated in the model for future use i.e., in a continual learning type of setup. The new information might be useful for future feature revelations. Therefore, incorporating it in the base model or the model $\psi$ will be useful as otherwise any new information received after the training of the base model (say the base P-VAE) will be lost to future instance of new feature introductions. This seems to be a very practical requirement. Is there a way to achieve this without having a big mitigating impact on speed and efficiency?

It does seem that the focus is on an unsupervised setting (as P-VAE) was used. How will the CHN be used for classification or other types of supervised learning tasks.

---

> ### Author Response · Authors · 2020-11-24
> **Response**
>
> Thank you for your review, please find our responses below:
>
> •	what does it mean to have a fixed point for each k on the x-axis ?
>
> The purpose of Figure 5 is to demonstrate the performance of CHN and baseline methods on test inputs with varied number of observed features. Therefore in Figure 5, the context set size $k_n$ is fixed to a value $k$ for each point on the x axis. That is to say, we use a context set of $k_n=k$ observations as input to each method for each new feature. This plot clearly shows the benefits of CHN in low observation regime and the trend of performance by varying the number of observations. During training, $k_n$ is sampled uniformly from Uniform [0,32] to ensure that the CHN is able to generate good parameters for a range of values of k.
>
> •	How is the new feature incorporated in the model for future use i.e., in a continual learning type of setup.
>
> It would certainly be possible to use the CHN to predict e.g. the embedding weights associated with a new feature in the PVAE encoder, so that if this feature is included in future data points the model is able to encode it in a useful manner without requiring retraining of the model. However, we did not consider encoding additional features in our experiments.
>
> •	It does seem that the focus is on an unsupervised setting (as P-VAE) was used. How will the CHN be used for classification or other types of supervised learning tasks.
>
> The CHN is intended to be used for supervised tasks and relies on a supervised training loss – the P-VAE model is used as a flexible example model that can make predictions for missing features, where the identity of the missing features varies between each data point as in e.g. recommender system applications. In the original paper, the P-VAE is applied to supervised learning settings when the missing values of the input data are considered targets. For example, the medical dataset can be viewed as a binary classification task, where each of the unobserved pathophysiologies is considered a prediction target. This can be extended simply to multi-class classification through use of a categorical likelihood term for the corresponding target dimensions.

---

### Official Review · AnonReviewer2 · 2020-10-24
**Review (update)**

**Rating:** 5
**Confidence:** 3

**Review:**

The paper proposes a few-shot meta-learning method for recommender system that uses a new feature's meta-information and observed samples for the features to predict the network weights for predicting the feature value from other features. The paper focuses on the cold-start problem where few samples with a new feature observed is available. The method outperforms a wide range of baselines on MovieLens-1M, a medical synthetic dataset, and a e-learning dataset.

Disclaimer: I am not familiar with recommender systems.

Pros:
- Methods straightforward and easy to understand, and does not have much ad-hoc design choices that are hard to validate.
- The experiments are relatively thorough. A lot of baselines.

Cons
- Novelty is limited -- a mixture of meta-learning and content-based method. There are also existing few-shot continual learning papers available (e.g. https://openaccess.thecvf.com/content_CVPR_2020/papers/Tao_Few-Shot_Class-Incremental_Learning_CVPR_2020_paper.pdf), so the motivation needs to consider differences from them.
- Experiments does not do external comparisons with other recommender systems that deal with cold starting other than its own baselines.
- Despite the large amount of baselines, ablation still lacks: (1) Train from random, but instead of training for a fixed number of epochs, just train an SVM or linear regression until convergence. This is what people would do as a baseline. (2) not important but it would be nice to ablate the CHN by taking out metadata and $C_n$ from input.
- Not clear if the set of datasets is persuasive. One is a synthetic dataset. One recommendation system and one grading dataset.
- There is no Appendix despite referencing it.

Minor issues:
- Not clear how the "adapting to new features" is different from meta-learning's adapting to new tasks, since they also use old tasks to inform new tasks. The abstract seems to suggest the method can take new features as input incrementally, which is a little confusing.
- Overclaim at the end of Section 2.2: it is not O(1) if you have to do distributed computing. If you allow distributed computing, NP=P.
- It's hard to tell why MAML is similar to CHN in one dataset but underperforms drastically in the e-learning dataset.


Post-rebuttal:
I appreciate the additional ablation study, but unfortunately the results did not strengthen the paper's distinction from related work. The explanation of the motives and related work comparisons only clarified differences between this paper and prior work that are either inherent to the task being addressed, or contribution unsupported by experiments. Unless the AC agrees with the authors that the paper is acceptable even without external comparisons (despite being a merge of two lines of work), I will not be changing my score.

---

> ### Author Response · Authors · 2020-11-24
> **Response**
>
> Thank you for your review, please find our responses below:
>
> •	.. existing few-shot continual learning papers available...
>
> Our approach differs from existing meta-learning approaches in that CHN explicitly utilizes a data point’s task labels for previous tasks (i.e. existing features in the model) when adapting to a new task. On the other hand, existing meta-learning methods for image classification applications typically treat each task independently and do NOT use previous task labels in an explicit manner. For instance, if we are given an image of a red car and a new class to be learned is “four wheeled vehicles”, most meta-learning approaches would not make explicit use of the fact that this data point may have been labelled as “car” or “red” in previous tasks, instead relying on intermediate feature representations from a classifier or adapting parameters. We instead sought a method for multi-task learning that could make explicit use of all such past labels for all data points in the context set, as these labels effective constitute the entirety of the data point in collaborative filtering-based approaches to recommender systems.
>
> •	external comparisons with other recommender systems
>
> In our experiments we aimed to assess the efficacy of CHNs versus a range of baselines when applied to a consistent model, in order to ensure a fair comparison between the methods: as such, we were more interested in assessing the relative ordering of performance between the methods in order to highlight the efficacy of CHNs, than the absolute performance of the overall system. To be clear, our contribution is intended to be the CHN rather than the P-VAE + CHN system, and the P-VAE is provided as an exemplar model. We do not contest that there are likely models in the recommender system literature which will yield better absolute performance, and hope that applying CHNs to such models may be a promising direction for future work in recommender systems.
>
> •	It would be nice to ablate the CHN by taking out metadata and Cn from input.
>
> An ablation of the performance of the CHN when $\mathcal{C}_n$ is removed is effectively provided by the performance plots in the paper (e.g. Figure 5) by reading the performance at $k=0$, as this corresponds to an empty context set. We have additionally included an ablation for the use of metadata on MovieLens-1M in Appendix C.5. This shows a very small effect, which we believe is due to the feature metadata for MovieLens-1M being uninformative relative to the data points themselves. We will try to provide the additional baselines described if time permits.
>
> •	Not clear if the set of datasets is persuasive.
>
> We tried to cover three realistic application scenarios for CHNs: both the recommender system and E-learning datasets are large (>1 million observations), with the education dataset one of the largest available. While a simulated dataset, the medical dataset was designed to preserve the properties of a corresponding real-world dataset while side-stepping the associated privacy concerns, and so we believe it represents a real world setting for medical applications.
>
> •	There is no Appendix despite referencing it.
>
> Thank you for bringing this to our attention: the appendices were included as supplementary material, these have been added to the main paper in the revised version.
>
> •	Not clear how the "adapting to new features" is different from meta-learning's adapting to new tasks, since they also use old tasks to inform new tasks. The abstract seems to suggest the method can take new features as input incrementally, which is a little confusing.
>
> See response to first point. We want to learn a method that can explicitly reason about each data point in the context set’s labels on other tasks, rather than treating each task as independent. . For example, the ratings of different movies from the same user are not independent tasks, but classifying between MNIST digits 1 and 2 and between digits 3 and 4 are independent tasks.
>
> •	Overclaim at the end of Section 2.2: it is not O(1) if you have to do distributed computing. If you allow distributed computing, NP=P.
>
> Thank you for bringing this to our attention, we have updated the manuscript accordingly.
>
> •	It's hard to tell why MAML is similar to CHN in one dataset but underperforms drastically in the e-learning dataset.
>
> Appendix B includes a detailed description of the protocol used for MAML, and it also includes figures for performance across a range of fine-tuning epoch numbers. Since MAML fine-tunes from a single shared head initialisation, it may be that some datasets require a greater degree of task-specific adaptation of the head parameters than others.

---

> > ### Comment · AnonReviewer2 · 2020-11-25
> > **Areas of disagreement with the rebuttal**
> >
> > > Our approach differs from existing meta-learning... For instance, if we are given an image of a red car and a new class to be learned is “four wheeled vehicles”, most meta-learning approaches would not make explicit use of the fact that this data point may have been labelled as “car” or “red” in previous tasks, instead relying on intermediate feature representations from a classifier or adapting parameters.
> >
> > I see the difference, but I see no problem relying on intermediate feature representations -- for example, in prototypical networks for few-shot learning, the embedding can be learned in such a way that the average representation of a class can be used as its classifier. Images for "four wheeled vehicles" and images for "car" and "red" would have similarities in their features, and the prototypical networks can use past task information as well. The only difference from these methods is then the introduction of explicit meta information, which I was asking for ablation studies. Unfortunately it does not show a significant difference ("...ablation for the use of metadata on MovieLens-1M in Appendix C.5. This shows a very small effect..."). Therefore, a reader would like to know what is different in practice about this method vs other few-shot learning papers, which the experiments do not provide.
> >
> > Also regarding using other tasks' info: it is especially hard to NOT directly use other tasks in a recommender system, seeing that prior tasks are literally used as features as the input, so using meta-learning / few-shot learning on recommender systems almost guarantees directly using info from other tasks, so I am skeptical about claiming this as a separate difference than just applying those methods in recommender systems.
> >
> > > In our experiments we aimed to assess the efficacy of CHNs versus a range of baselines when applied to a consistent model, in order to ensure a fair comparison between the methods...
> >
> > If the paper only wants to prove their additions to be effective, I see much less impact because inferring classifiers for few-shot learning is proven to work in prior prototypical few-shot learning work, and adding meta-info to improve performance sounds straightforward.
> >
> > > Appendix
> >
> > Terribly sorry about that! I was not aware that openreview allows supplemental material.
> >
> > > Despite the large amount of baselines, ablation still lacks: (1) Train from random...
> >
> > This critique is missed in the rebuttal.

---

### Official Review · AnonReviewer1 · 2020-10-28
**Experimental setup leaves something to be desired**

**Rating:** 5
**Confidence:** 3

**Review:**

The paper proposes Contextual HyperNetworks (CHNs) as an auxiliary model to generate parameters from existing data, and observations and other metadata associated with new feature to address cold start problem of new feature. Besides, it doesn’t need either re-train or fine-tune at prediction time. The CHN is applied to P-VAE and some experimental results are provided to demonstrate its effectiveness in some application, i.e., recommender system, e-learning and healthcare tasks.

The motivation is clear and reasonable, and the proposed method with hyper-network is pretty interesting and attractive. However, experimental section has large improvement room.

 Pros:

* The paper proposes an interesting direction to use hyper-network to solve cold-start problems in some critical tasks, such as recommender systems and etc.
* The logic of paper is clear and easy to follow.

Cons:

* The paper only illustrates how to apply CHNs to P-VAE, however, it’s better also to illustrate how to apply CHNs to other pretty common techniques in recommender system field (or e-learning or others), such as deep learning-based collaborative filtering methods. If there is no more one application, it is difficult to demonstrate the proposed method is generalizable enough to other models, including how easy it could be extended and how effectiveness it could have after extension.
* The paper only shows the advantage of prediction time, but it doesn’t discuss a lot on the training latency. It’s better to also discuss this in the paper and compare with other methods.
* The experiment setup is not strong enough to demonstrate the effectiveness of the proposed methods, for example,
    * Lack of major important baselines and studies. In the paper, only several extension on how to handle new features on top of P-VAE is given in the comparison. However, it is unclear how it performs with other methods which target at solving cold-start problem. In other others, current experiments cannot provide evidence to show it outperforms other methods which could be used to address cold start problem.
    * Fairness regarding to no meta information is incorporated in the baselines. CHNs is leverage additional meta information, however, other baselines don’t use this kind of information.
    * Limited evaluation metics. Each task only either uses RMSE or AUC. So it lacks evidence to show whether the proposed method could outperform others in different validation metrics instead of bias.

---

> ### Author Response · Authors · 2020-11-24
> **Response**
>
> Thank you for your review, please find our responses below:
>
> •	...illustrate how to apply CHNs to other pretty common techniques in recommender system field..
>
> The current formulation is applicable to any deep learning-based model that can be extended to additional features, by taking some intermediate representation of the input features and using these to represent the context set associated with predicting a new feature.  Firstly, P-VAE represents generic auto encoder models in recommender systems. For example, any autoencoder-based method taking user histories as input, such as (Liang et al, 2018) or (Sedhain et al., 2015) permit a near-identical application of P-VAEs. Thus, applying our method is very straightforward. Secondly, other methods such as deep matrix factorization (Xue et.al 2017) can benefit from our methods more as both users and items are treated as fixed sized vector in the original setting. We can apply our method to both the user embedding learning part of the network as well as item embedding part of network in the same way as how we used in combination with P-VAEs.
>
> •	. the training latency...
> Thank you for highlighting this: description of training times for CHNs was included in an appendix as part of the supplementary material for the original submission and may have been missed. To clarify this, we have moved this information to a new Timing Experiments subsection (4.5).
>
> •	Lack  baselines and studies....
> We aimed to provide a wide range of baselines that could be applied in this setting to solve the cold start problem, including item-based approaches such as nearest neighbours, content-based approaches such as averaging the parameters matching the new feature’s metadata, and meta-learning approaches such as MAML. We aimed to augment existing meta-learning approaches to be able to explicitly utilize information about previous tasks (i.e. existing features in the model) when adapting to a new task, rather than treating each task independently, and felt that many existing meta-learning methods would not apply in this setting. If you have any suggestions for specific baselines that you feel should be given a comparison, we would be happy to consider them.
>
> •	...no meta information is incorporated in the baselines...
> We believe our experimental setting is fair to baselines: the mean head parameters baseline used for MovieLens-1M and the E-learning dataset also utilises metadata. This is done by taking the mean of the trained head parameters for training set features only over features whose metadata categories are a subset of those of the new feature under consideration. We have included a comparison of performance with/without metadata for both CHNs and mean head parameters in Appendix C.5, Figure 10.
>
> •	Limited evaluation metics. Each task only either uses RMSE or AUC.
>
> For the experiments on the Neuropathic Pain and E-learning datasets, we have additionally included the RMSE values in Appendix C.4, Figure 9 in order to provide another evaluation metric for these tasks, and we see that the CHN remains competitive when assessed by this metric. AUROC was chosen in place of accuracy for the main text in order to better accommodate imbalanced classes in the two classification tasks. The fact that CHN performs better/competitive in both RMSE and AUROC shows our approach is indeed better than baseline approaches across a range of datasets for different applications.
>
> References:
> Liang, Dawen, et al. "Variational autoencoders for collaborative filtering." Proceedings of the 2018 World Wide Web Conference. 2018.
> Sedhain, Suvash, et al. "Autorec: Autoencoders meet collaborative filtering." Proceedings of the 24th international conference on World Wide Web. 2015.
> MLA, Xue, Hong-Jian, et al. "Deep Matrix Factorization Models for Recommender Systems." IJCAI. Vol. 17. 2017.

---

### Official Review · AnonReviewer4 · 2020-10-28
**An okay submission but with limited novelty.**

**Rating:** 5
**Confidence:** 4

**Review:**

This submission focuses on the cold start problem of new entities (new items in a recommender system, new treatments in a medical application, etc.). It combines the strengths of the *relations* between a new entity and the existing entities, and the *content* features of the new entity, by fusing the two kinds of information into a neural network that outputs the estimated representation of the new entity. The proposed method outperforms several intuitive naïve strategies as well as MAML.

Pros:
- The writing is clear.
- Good reproducibility. Details, including hyperparameters, are listed in the appendix.

Cons:
1. Similar models -- models that take the graph topology around a new node as well as the node’s feature as input and produces the node’s embedding without any extra training/finetuning step -- have long existed in the literature. For example, one can google “inductive learning + graph embedding”, “out-of-sample extension + graph embedding”, etc., and find plenty of related works. Not to mention that the nowadays (over-)popular graph neural networks naturally support this.
2. It is unclear how this work improves upon Vartak 2017. Vartak 2017 views the cold-start problem in recommender systems as a meta-learning problem as well. It also combines the user rating data and the item’s content features. What’s new when compared to Vartak 2017, and how well the proposed method outperforms Vartak 2017 empirically?

---

> ### Author Response · Authors · 2020-11-24
> **Response**
>
> Thank for your review, please find our responses below:
>
> •	...models that take the graph topology..
>
> Thank you for pointing out the GNN work which can address a similar issue. We have added a paragraph in the end of related work discussing these related works. In general, it is indeed true that GNN based methods naturally can encode new nodes. However, there is a large range of applications where GNN not be suitable, which can either be due to its high computational or memory requirement, or the problem setting may not be easily formulated as graphs. Our work addresses more generic deep learning models beyond GNN, for example vector based neural networks as we used in the experiments.
>
> •	... Vartak 2017. Vartak 2017 views the cold-start problem in recommender systems as a meta-learning ...
>
> Thanks for highlighting the relationship between our work and (Vartak et al., 2017) , we have extended our discussion of this work in the revised submission. We believe that our work is substantially different from (Vartak et al., 2017). Whereas this method learns either the weights of a linear classifier, or solely the biases of a non-linear classifier, whereas we learn both. Their method can only be applied to classification problems as it relies on a class representative embedding R^c_j, whereas our method is applicable to both classification and regression problems. The architectures are very different in general, with the “main model” in (Vartak et al., 2017) taking an item history embedding and then passing it through a network with weights generated by the user history, whereas we input the user history and pass it through a network whose final layer has weights generated by the item history. We view our main advantage over this method as our ability to augment a much larger primary model with a small number of new adapted parameters, as opposed to generating the whole network as they do, allowing us to keep the computational costs of adaptation small without sacrificing model capacity by using a smaller model or a factorization of the weights.

---

### Author Response · Authors · 2020-11-24
**Revised version**

We thank the reviewers for their reviews and helpful comments. We have uploaded a revised version of the submission. Major updates to the main text are highlighted in red. The major edits include:
1.	Inclusion of appendices in main submission rather than as separate supplementary material
2.	Expanded related work to include discussions on suggested papers.
3.	Moved the CHN training time information (originally in the appendix) to a new “Timing Experiments” subsection (4.5) in the main text.
4.	Updated appendix to include new results:
a.	An ablation on fine-tuning the new heads on the context set of meta test inputs, with the new heads initialised either randomly or using CHN (Appendix C.3). The results show that CHN initialisation is significantly better than training from a random initialisation for a range of context set sizes.
b.	The RMSE performance of CHN & baseline methods on the Neuropathic Pain and E-learning datasets (Appendix C.4). Results show that both RMSE and AUC metrics consistently rate CHN performance as highly competitive.
c.	An ablation study on the usage of meta-data in MovieLens-1M experiments (Appendix C.5).

We would also like to briefly clarify the main contribution of the paper. Our goal is to enable fast adaptation of a machine learning model when new output features are added to augment the originally observed data. This is different from typical meta-learning/few-shot learning setting where the new task is typically treated independently from previous tasks. Instead, the proposed CHN explicitly utilizes previous information for each datapoint to predict the newly added features in an amortised manner. We envision that CHN can be applied to generic machine learning models, and in the paper, we used the P-VAE as an exemplar model to carry out our empirical evaluations. Our experiments aim to cover a wide range of applications (recommender systems, E-learning, medical diagnosis) that this new feature adaptation task is relevant, and our results showed that CHN performs better than/competitively with all considered baselines on these applications.

---

### Decision · Program_Chairs · 2021-01-07
**Final Decision**

**Decision:**

Reject

**Comment:**

This work presents a straightforward and easy to understand method for using hypernetworks to adapt existing models to be able to increase their output space. The method itself is also interesting and is detailed enough for reproducibility. However, the experiments and results should be improved by expanding the demonstration of CHNs beyond the narrow P-VAE application and comparing against relevant baselines in the recommendation system literature.

Pros
- Clear writing.
- Detailed hyperparameters to aid reproducibility.
- Straightforward model.

Cons
- Lack of sufficient comparison to related work, especially to existing recommendation systems that handle the cold-start issue and to Vartak, 2017.
- Limited results that only demonstrate application to P-VAE meaning it's still unknown if CHNs work well with other models. The result on the synthetic dataset is also less persuasive.
- Lack of sufficient ablations, i.e. training a SVM/linear regression model until convergence.